

# Detailed Examination of Upper Troposphere Lower Stratosphere Composition Change from DCOTSS Airborne Observations of Active Convection on 31 May 2022

Andrea E. Gordon[1], Cameron R. Homeyer[1], Jessica B. Smith[2], Rei Ueyama[3], Jonathan M. Dean-Day[4], Elliot L. Atlas[5], Kate Smith[5], Jasna V. Pittman[2], David S. Sayres[2], David M. Wilmouth[2], Apoorva Pandey[2], Jason M. St. Clair[6,7], Thomas F. Hansico[6], Jennifer Hare[2], Reem A. Hannun[8], Steven Wofsy[2], Bruce C. Daube[2], and Stephen Donnelly[3]

[1]School of Meteorology, University of Oklahoma, Norman, OK, USA
[2]Harvard John A. Paulson School of Engineering and Applied Sciences, Harvard University, Cambridge, Massachusetts, USA
[3]NASA Ames Research Center, Moffett Field, California, USA
[4]Bay Area Environmental Research Institute, Moffett Field, California, USA
[5]Rosenstiel School of Marine, Atmospheric, and Earth Science, University of Miami, Miami, Florida, USA
[6]NASA Goddard Space Flight Center, Greenbelt, Maryland, USA
[7]University of Maryland Baltimore County, Baltimore, Maryland, USA
[8]Department of Geology & Environmental Science, University of Pittsburgh, Pittsburgh, Pennsylvania, USA

**Correspondence:** Andrea E. Gordon (agordon@ou.edu)

**Abstract.** Tropopause-overshooting convection in the midlatitudes provides a rapid transport pathway for air from the lower troposphere to reach the upper troposphere and lower stratosphere (UTLS), and can result in the formation of above-anvil cirrus plumes (AACPs) that significantly hydrate the stratosphere. Such UTLS composition changes alter the radiation budget and impact climate. Novel in situ observations from the NASA Dynamics and Chemistry of the Summer Stratosphere (DCOTSS)

field campaign are used in this study to examine UTLS impacts from AACP-generating overshooting convection. Namely, a research flight on 31 May 2022 sampled active convection over the state of Oklahoma for more than three hours with the NASA ER-2 high-altitude research aircraft. An AACP was bisected during this flight, providing the first such extensive in situ sampling of this phenomenon. The convective observations reveal pronounced changes in air mass composition and stratospheric hydration up to altitudes of 2.3 km above the tropopause and concentrations more than double background levels.

Unique dynamic and trace gas signatures were found within the AACP, including enhanced vertical mixing near the AACP edge and a positive correlation between water vapor and ozone. Moreover, the water vapor enhancement within the AACP was found to be limited to the saturation mixing ratio of the low temperature overshoot and AACP air. Comparison with all remaining DCOTSS flights demonstrates that the 31 May 2022 flight had some of the largest tropospheric tracer and water vapor perturbations in the stratosphere and within the AACP.





## 1 Introduction

The tropopause is often viewed as a sharp boundary separating the troposphere and stratosphere. However, dynamic processes routinely impact the tropopause region and modify the characteristics of a variably deep layer that is commonly referred to as the upper troposphere and lower stratosphere (UTLS). The most rapid of these processes, tropopause-overshooting convection

(hereafter "overshooting convection"), facilitates transport of air from the lower troposphere to the UTLS in minutes to hours and can enable irreversible mixing between these layers, known as Stratosphere-Troposphere Exchange (STE). Transport and corresponding STE from overshooting convection alters the composition of the UTLS, modifying greenhouse gases such as water vapor and ozone (Holton et al., 1995; Stohl et al., 2003). This can impact the radiation budget and climate, as the radiation budget is especially sensitive to composition changes within the UTLS. For example, the radiative forcing by water vapor in

the UTLS is most sensitive to changes in the extratropical lowermost stratosphere (Solomon et al., 2010; Banerjee et al., 2019), while that for ozone is most sensitive to changes in the upper troposphere (Lacis et al., 1990). Overshooting convection is also commonly severe, producing hazardous weather including hail, tornadoes, and strong winds (Fujita, 1974; Brunner et al., 2007; Bedka, 2011; Dworak et al., 2012; Bedka et al., 2015, 2018; Marion et al., 2019; Sandmæl et al., 2019). To date, much of what we know about the impacts of overshooting convection on the UTLS is based on remotely sensed observations and numerical

modeling studies. Few airborne field campaigns have extensively sampled (more than a few minutes) tropopause-overshooting convection, resulting in limited in situ observations of these storms (Smith et al., 2017; Homeyer et al., 2014b).

Tropopause-overshooting convection occurs worldwide and is most frequent over land in the midlatitudes, especially in North and South America (Liu and Liu, 2016; Liu et al., 2020). The United States, specifically the Great Plains region, has been recognized as a unique hotspot of deep and frequent overshooting based on satellite and radar observations (Solomon

et al., 2016; Cooney et al., 2018; Homeyer and Bowman, 2021). Overshooting convection in the United States is most frequent in the warm season (March-August), with notable annual and diurnal cycles. Approximately half of overshooting convection in the United States reaches the stratospheric overworld (potential temperature ($\theta$) > 380 K), where the impacts from these events on the stratosphere can last weeks to months as this air will take longer to descend back to the troposphere in the midlatitudes (Holton et al., 1995). Additionally, air that reaches the stratospheric overworld can be isentropically transported toward the

tropics where it may enter the the upward branch of the Brewer-Dobson circulation, further extending its stratospheric lifetime (Chang et al., 2023). Despite not being studied explicitly, overshooting convection frequency will likely increase in the future as favorable environments for severe thunderstorms are projected to increase with a warming climate (Del Genio et al., 2007; Trapp et al., 2007; Lepore et al., 2021).

A key storm feature commonly associated with midlatitude tropopause-overshooting convection is the above-anvil cirrus

plume (AACP). AACPs are cirrus clouds that reside above the anvil within the stratosphere, and are commonly identified in visible and infrared (IR) satellite imagery (Wang, 2003; Luderer et al., 2007; Wang et al., 2016; Homeyer et al., 2017). When AACPs appear warm relative to the underlying anvil cloud in IR imagery, this indicates stratospheric injection of cloud ice and subsequent hydration via sublimation (Murillo and Homeyer, 2022). These plumes are known to be driven by gravity wave breaking near and within the overshooting top in environments with strong storm-relative winds in the UTLS, which



is associated with the establishment of a hydraulic jump (Wang, 2003; Luderer et al., 2007; Wang et al., 2016; Homeyer et al., 2017; O'Neill et al., 2021). While our appreciation for the impact that various storm and environmental characteristics (including whether an AACP is produced) have on the amount of overshooting and cross-tropopause transport is increasing, most prior observational work does not consider such characteristics when evaluating the UTLS impacts associated with these storms.

UTLS composition impacts from overshooting convection have been explored broadly via satellite observations (Randel et al., 2012; Schwartz et al., 2013; Smith et al., 2017; Werner et al., 2020; Tinney and Homeyer, 2021) and specifically using observations from airborne field campaigns (Poulida et al., 1996; Fischer et al., 2003; Fromm and Servranckx, 2003; Hegglin et al., 2004; Hanisco et al., 2007; Homeyer et al., 2011; Anderson et al., 2012; Homeyer et al., 2014b; Pan et al., 2014; Herman et al., 2017; Smith et al., 2017). These studies have shown that water vapor in the UTLS is enhanced through both

sublimation of ice and air mass transport, and that water vapor and additional tropospheric and stratospheric trace gases have been observed to be modified up to 4 km above the tropopause. Stratospheric hydration from convection has been encountered in a few prior field campaigns including: the SCOUT-AMMA aircraft campaign over Africa (Khaykin et al., 2009), the Aura Validation Experiment (AVE) aircraft campaign over the United States (Hanisco et al., 2007; Anderson et al., 2012; Smith et al., 2017), the Studies of Emissions, Atmospheric Composition, Clouds and Climate Coupling by Regional Surveys (SEAC4RS)

aircraft campaign over the United States (Smith et al., 2017), and the Deep Convective Clouds and Chemistry (DC3) field campaign over the United States (Homeyer et al., 2014b; Pan et al., 2014). The majority of convective encounters within these field campaigns was serendipitous rather than targeted. The SEAC4RS campaign targeted convective outflow from an AACP-producing storm that occurred ∼20 hours prior to flight, detailed in Smith et al. (2017). In situ data were used to estimate that between 6.6 and 13.5 kilotonnes of water vapor was irreversibly delivered to the lower stratosphere from this convective

system. Most airborne observations of overshooting convection and UTLS impacts are associated with recent (less than 36 hours between overshooting and sampling) or aged convection (longer than 36 hours between overshooting and convection) rather than active convection (measurements in or near an active convective complex with co-located cloud observations, which can range from minutes to a few hours old), especially within the lower stratosphere. The DC3 field campaign had active convection targeting, which occurred primarily in the upper troposphere. The G5 and DC-8 aircraft used during DC3, however,

did encounter convectively injected $H_2O$ at altitudes up to 1-2 km above the tropopause near active overshooting convection during a flight on 20 May 2012, detailed in Homeyer et al. (2014b). Water vapor enhancements exceeded 200 ppmv above background concentrations in the lower stratosphere, with evidence of extensive mixing between troposphere and stratosphere air within this broad $H_2O$ enhancement. Additionally, an AACP was briefly encountered within this flight, measured at a distant range from the parent overshoot for approximately 30 seconds.

As past observations of overshoot-influenced air are limited, much of the current understanding of the impact of overshooting convection on the UTLS has been gleaned from numerical model simulations. Simulations of overshooting convection suggest the amount of overshooting and cross-tropopause transport is sensitive to storm mode (Mesoscale Convective System or MCS versus a discrete supercell) and the characteristics of the lower stratospheric environment, such as stability and the strength of storm-relative winds (Mullendore et al., 2005; Bigelbach et al., 2014; Homeyer et al., 2014a; Starzec et al., 2020; O'Neill



et al., 2021; Gordon and Homeyer, 2022). The amount of STE from overshooting convection also appears to depend on the height of the tropopause and, more generally, UTLS temperature (Phoenix and Homeyer, 2021). A lower UTLS temperature is an important constraint for stratospheric hydration, while a higher tropopause also impacts air mass transport as it is harder to reach since it allows for greater cumulative diluting effects on air within the convective updraft that result from entrainment of environmental (free troposphere) air. Within these known sensitivities, multiple simulations have shown that water vapor en-

hancements have appeared to reach higher altitudes than passive tropospheric tracers (Homeyer, 2015; Phoenix and Homeyer, 2021; Gordon and Homeyer, 2022). Furthermore, recent numerical modeling studies have identified the unique influence of AACP-producing storms on UTLS composition (O'Neill et al., 2021; Gordon and Homeyer, 2022). Specifically, troposphere-to-stratosphere (TST) transport is greater in AACP-producing storms. Additionally, AACP-producing storms feature enhanced downward transport of overworld air to the lowermost stratosphere (the portion below 380-K potential temperature) compared

to non-AACP storms (Gordon and Homeyer, 2022). Many of these modeling results have yet to be verified by observations, including water vapor enhancements routinely reaching greater altitudes than tropospheric tracers and the apparent downward mixing of overworld air to the lowermost stratosphere associated with AACP-producing storms.

The recently completed NASA Dynamics and Chemistry of the Summer Stratosphere (DCOTSS) field campaign provides unprecedented observations of tropopause-overshooting convection that have exceeded all prior records of stratospheric depth

(Homeyer et al., 2023). The primary goal of the DCOTSS mission was to improve understanding of the dynamical and chemical processes that influence the composition of the extratropical summertime stratosphere, with several core science questions including: (a) How much tropospheric air and water is irreversibly injected into the stratosphere by convection?, and (b) What dynamical mechanisms lead to the irreversible injection of material into the stratosphere by convective storms? DCOTSS completed at least 18 flights sampling material from overshooting convection, but it is currently uncertain how many of these

flights sampled material from AACP-producing storms (versus non-AACP storms). Multiple active convection flights were completed, but only one flight (on 31 May 2022) specifically targeted and repeatedly sampled an actively-generated AACP as it was deemed safe to do so. In this study, we present in situ airborne observations of this event. The measurements obtained throughout this flight by the versatile DCOTSS instrument payload provide unique insight into UTLS composition changes associated with the storm. We seek to utilize these observations to address how the composition of the UTLS is altered by

an AACP-producing storm and compare these observations with results from recent modeling studies. Specifically, we aim to address the following questions: (i) How is the UTLS composition altered near/during active overshooting convection and AACPs?, and (ii) What dynamical/physical processes could contribute to these changes in UTLS composition? We use airborne observations of trace gases and meteorological parameters to investigate composition changes and to constrain transport and mixing processes within the UTLS.





## 2 Data and Methods

### 2.1 DCOTSS Overview

The DCOTSS field campaign used the National Aeronautics and Space Administration (NASA) ER-2 high-altitude research aircraft to investigate the impact of intense convection on the summertime stratosphere over North America from June to August 2021 and May to July 2022. The focus was on sampling air from active, recent, and aged overshooting events over the United States, but science flights also sampled pyrocumulonimbus smoke, outflow from storms occurring over the Sierra Madre Occidental in Mexico, and surveyed the large-scale structure of the North American Monsoon Anticyclone. DCOTSS was based primarily in Salina, Kansas and the ER-2 flew 29 science flights with research-quality data (depending on the instrument) during the campaign: 14 in 2021 and 15 in 2022. Active convection was sampled during three flights (31 May 2022, 08 June 2022, and 24 June 2022), with the most extensive sampling of active convection during the 31 May 2022 flight. The 31 May 2022 flight sampled multiple convective enhancements (both inside and outside of cloud material) throughout the flight, featuring a bisect and profile of an AACP. The DCOTSS instrument payload consists of twelve instruments and was designed to provide myriad in situ measurements of trace species and meteorological parameters in the UTLS necessary to evaluate transport, mixing, and the net chemical impact of summertime convection on the UTLS.

### 2.2 DCOTSS Observations

In our analysis we use multiple select trace constituents that are useful for diagnosing convective transport within the UTLS. These include water vapor ($H_2O$), ozone ($O_3$), carbon monoxide (CO), methane ($CH_4$), total water (vapor + ice), and several additional trace species with anthropogenic sources at the surface. Harvard Water Vapor (HWV) measures ambient $H_2O$ mixing ratios at 1 Hz. The HWV instrument is composed of the Harvard Lyman-$\alpha$ photo-fragment fluorescence instrument (LyA) and the Harvard Herriot Hygrometer (HHH), a tunable diode laser direct absorption instrument. Only the HHH observations are used in this study. The measurements have an accuracy of 5-10% and a precision better than 0.1 ppmv (Sargent et al., 2013). The Rapid OZone Experiment (ROZE) is a cavity-enhanced ultraviolet absorption instrument that measures in situ $O_3$ by direct absorption, obtaining measurements at 1 Hz with a precision of 1 ppbv in the stratosphere and an accuracy of 6% (Hannun et al., 2020). The Harvard University Picarro Cavity Ringdown Spectrometer (HUPCRS) is a modified G-2401m Picarro gas analyzer (Crosson, 2008) that measures $CO_2$, $CH_4$, and CO every ∼2.2 secs; $CH_4$ data are reported at 1 Hz with 0.20 ppbv uncertainty and CO data reported at 0.1 Hz with 3.20 ppbv uncertainty. The Water Isotopologues – Integrated Cavity-Output Spectrometer (WI-ICOS) measures total water (vapor + ice) and its major isotopologues at 1 Hz. WI-ICOS uses cavity enhanced absorption, and an isokinetic inlet and heaters are used to measure total water, which is the only measurement we use in this study. WI-ICOS total water measurements have an accuracy of 0.1 ppmv and an uncertainty of 10% (Sayres et al., 2009).

The Advanced Whole Air Sampler (AWAS) consists of 32 stainless-steel canisters mounted in the center belly pod of the ER-2. The canisters were typically filled on-demand to target samples from specific features during a research flight. The time for AWAS to collect a sample varies depending on altitude, between 20-200 s (with shorter times at lower altitudes). After each



flight, canisters are analyzed using gas chromatography with mass spectrometry, a flame ionization detector, and an electron capture detector to determine the mixing ratio of a variety of tracers. AWAS measures more than 20 constituents with varying
lifetimes and provides valuable measurements of short-lived and very short-lived tracers (Flocke et al., 1999; Schauffler et al., 1999). AWAS constituents of interest for the 31 May 2022 flight include ethane, ethyne, and propane. Ethane ($C_2H_6$) has a tropospheric lifetime of 52 days (Chelpon et al., 2021), and is primarily sourced from crude oil and natural gas production, with additional sources from biomass and fossil fuel burning. Propane ($C_3H_8$) is also primarily a product of crude oil and natural gas production, with additional sources from biomass and fossil fuel burning, and has a tropospheric lifetime of 9.4
days (Chelpon et al., 2021). Ethyne ($C_2H_2$) has a tropospheric lifetime of 9.8 days (Chelpon et al., 2021), and is primarily sourced from biomass and biofuel burning, but can also result from crude oil and natural gas production.

Meteorological parameters including temperature, potential temperature, horizontal winds, and vertical winds are examined to gain further insight regarding STE processes throughout the flight. The Meteorological Measurement System (MMS) provides calibrated, high resolution measurements of ambient meteorological parameters, including temperature, pressure, and the
3-dimensional wind vector, at 20 samples per second. The MMS potential temperature has an accuracy of 0.5-1.5 K, temperature has an accuracy of $\pm0.3$ K, horizontal and vertical winds have a combined accuracy of $\pm1$ ms$^{-1}$, and the GPS altitudes have an accuracy of 15 m (Scott et al., 1990). For contextual campaign-wide analysis, we leverage feature identifications outlined in Homeyer et al. (2023). Namely, convective versus non-convective observations are based on manual identification of water vapor enhancements above prevailing background conditions within the stratosphere that have been linked to
overshooting convection (identified with radar and satellite observations) via trajectory matching.

### 2.3 ERA5 Reanalysis

Version 5 of the European Centre for Medium-range Weather Forecasts (ECMWF) reanalysis, ERA5, is used for tropopause identification within this study (Hersbach et al., 2020). ERA5 consists of hourly global assimilations of the atmospheric state on 37 pressure levels and a 0.25° latitude-longitude grid. The ERA5 vertical profiles are linearly interpolated to a regular
200-m grid before the lapse-rate tropopause (LRT) definition (World Meteorological Organization, 1957) is applied, resulting in tropopause altitudes with uncertainties $\leq$250 m (e.g., Hoffmann and Spang (2022)). LRT altitudes are interpolated linearly in time and space to aircraft location as well as radar and satellite grids for analysis. Although ERA5 output is available on the full 137 model grid, those data yield minimal refinement compared to the analyses included here.

### 2.4 Radar Observations

Gridded NEXRAD WSR-88D Radar (GridRad) data version 4.2 is used for high resolution analyses of overshooting convection in the contiguous United States (Homeyer and Bowman, 2022). GridRad merges individual NEXRAD WSR-88D data onto a common three-dimensional grid, and is available at 10-minute intervals for this study. The grid has ∼0.02° latitude-longitude (∼2-km) resolution, with a domain spanning 24-50°N latitude and 235-294°E longitude. Altitude spacing of the GridRad data is 0.5-km at altitudes below 7 km above mean sea level (AMSL) and 1-km otherwise up to 22 km AMSL. The sole
variable from the GridRad volumes used for analysis in this study is the radar reflectivity at horizontal polarization ($Z_H$).



Tropopause-overshooting convection is identified as $Z_H$ = 15 dBz echo-top altitudes above the ERA5 LRT. Quality control metrics consistent with previous radar climatologies (Solomon et al., 2016; Cooney et al., 2018; Homeyer and Bowman, 2021) are applied, and GridRad echo-top altitudes are unbiased with an uncertainty of 1 km.

## 2.5 Satellite Observations

Imagery from the Geostationary Observing Earth Satellite (GOES) platform over North America is used to enable AACP identification and to complement radar observations for analysis of the 31 May 2022 flight. Here, we only use GOES-16 (GOES-East) imagery at 1-minute intervals over the domain of the flight. Namely, visible (VIS; channel 2) and infrared (IR; channel 13) satellite imagery from the GOES Advanced Baseline Imager (ABI) are used (Schmit et al., 2017). AACPs are commonly identifiable in VIS as an area with relatively smooth texture that casts a shadow on the underlying anvil, especially

at times approaching sunset when the solar inclination angle is high. In IR, AACPs representing stratospheric injection of cloud material can be identified by anomalously warm brightness temperatures compared to the storm anvil (Murillo and Homeyer, 2022).

## 3 Results

### 3.1 Storm Characteristics and Flight Overview

The 31 May 2022 flight was designed to target active and ongoing overshooting convection in western Oklahoma. On 31 May 2022, a cluster of storms formed in western Oklahoma and the Texas panhandle near 21:00 UTC. Overshooting convection began near 22:00 UTC on 31 May 2022 and dissipated near 6:30 UTC on 1 June 2022, leading to ~8.5 hours of sustained overshooting within this cluster of storms. Overshoots reached a maximum echo-top height of 19 km at 22:00 UTC, with persistent echo-top heights near 17-18 km for the majority of the storms' lifetimes (including the sampling timespan of the

31 May 2022 flight). Figure 1 shows radar reflectivity, echo-top heights, and IR & VIS satellite imagery of the storms for select times, including throughout the sampling time period (00:20 UTC through 02:10 UTC). Material from the overshoot and accompanying AACP in western Oklahoma (indicated by the yellow and cyan circles in Fig. 1, respectively) was targeted. This specific cell had sustained overshooting from 22:00 UTC on 31 May until it decayed near 01:00 UTC on 1 June (3 consecutive hours of overshooting), while the AACP remained until at least ~03:00 UTC on 1 June. The convectively influenced air from

the overshooting storm was advected to the east, while the storm that produced the AACP moved eastward more slowly. As a result, the ER-2 could safely sample the overshoot material downstream of the storm for an extended period of time. The 31 May 2022 flight was the first active convection flight of DCOTSS and the only flight to target an AACP.

Figure 2 shows the entire flight track of the 31 May 2022 flight colored by altitude and $H_2O$ measurements, with points of interest labeled. The flight began with takeoff at 22:59 UTC on 31 May 2022 and the ER-2 ascended to an altitude of 20

km while approaching the targeted storms in western Oklahoma, approximately 1-2 hours after their initial overshooting. The aircraft descended to the first level leg at an altitude of 14.5 km near point A at 00:20 UTC (Figs. 2 & 3a), and completed a





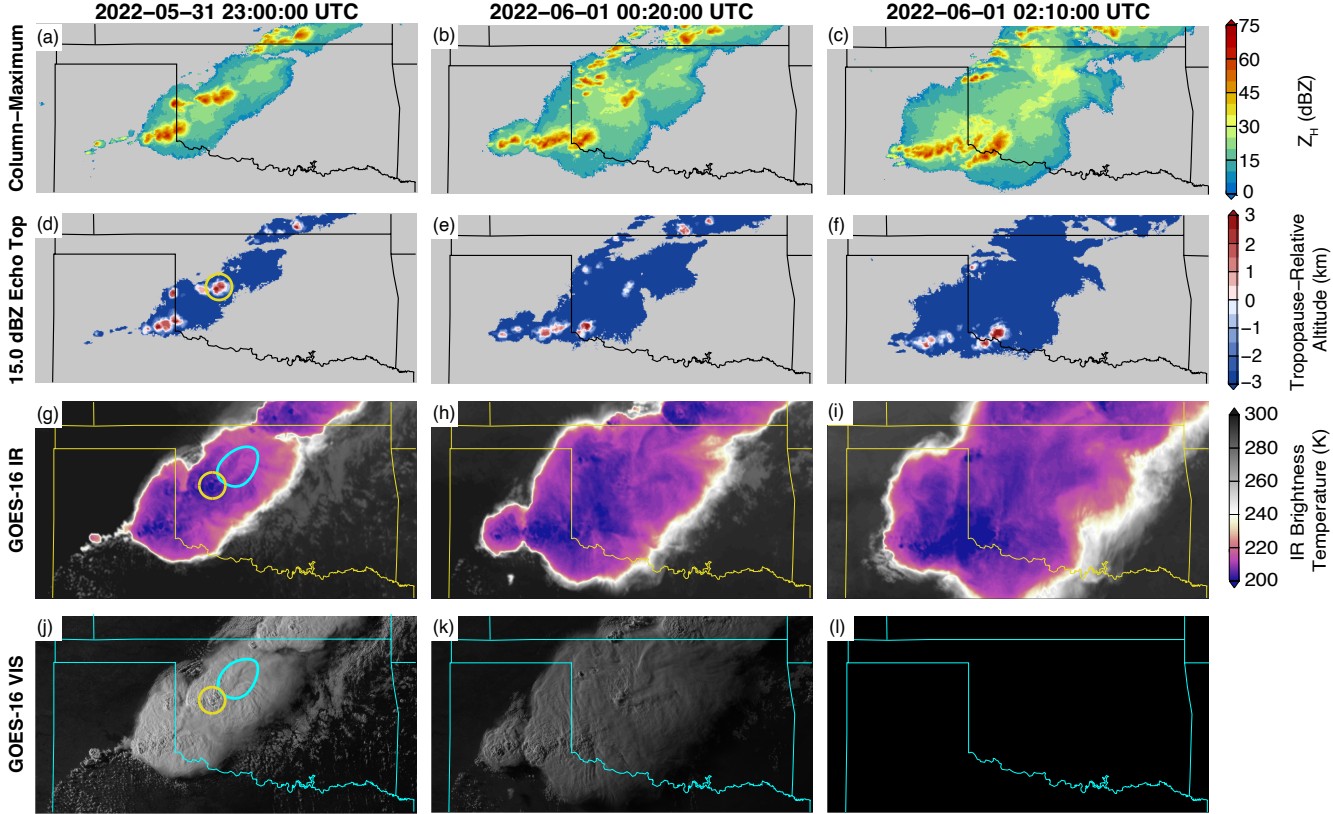

**Figure 1.** From top to bottom, GridRad radar reflectivity ($Z_H$; panels a–c), GridRad $Z_H$ = 15.0 dBZ tropopause-relative echo top heights (km; panels d–f), GOES-16 infrared imagery (panels g–i), and GOES-16 visible imagery (panels j–l) at: (left) 23:00 UTC on 31 May 2022 (takeoff), (middle) 00:20 UTC on 01 June 2022 (convective sampling begins), and (right) 02:10 UTC on 01 June 2022 (start of the AACP bisects). The targeted overshoot is annotated with a yellow circle, while the targeted AACP is annotated with a cyan oval.

series of stacked level legs between points P1 and P2, at altitudes of 14.5, 15, 14.75, 15.5, and 16.5 km (Fig. 2). The AACP of the targeted storm can be seen in ER-2 camera imagery from the level legs in Figs. 3a (facing forward) and 3b (facing right of the aircraft, towards the west). Beginning at 02:07 UTC (~3:10 hours of flight time), following the radar-indicated collapse of
the target cell near point P3 at approximately 01:00 UTC (seen in Fig. 1), the ER-2 was able to sample between points P1 and P3 at an altitude of 15.75 km, bisecting the AACP. The ER-2 image midway through the AACP bisect shows a layer of light distinguishing the AACP from the broader anvil below (Fig. 3c; facing right of the aircraft). The pilot encountered turbulence near point P3 and briefly ascended to 16.75 km before turning northeast and returning to 15.75 km, resulting in a fortuitous vertical profile through the AACP. Fig. 3d shows the AACP as the ER-2 ascends above it during the vertical profile near P3.
The camera was again facing to the right). As the ER-2 approached point P1 following the AACP bisect, it was routed back to Salina, KS at 02:48 UTC due to convection near the airport that could impact landing. The ER-2 ascended to an altitude of 20 km and returned to Salina, KS at 04:07 UTC, completing the 5 hour, 8 minute flight.



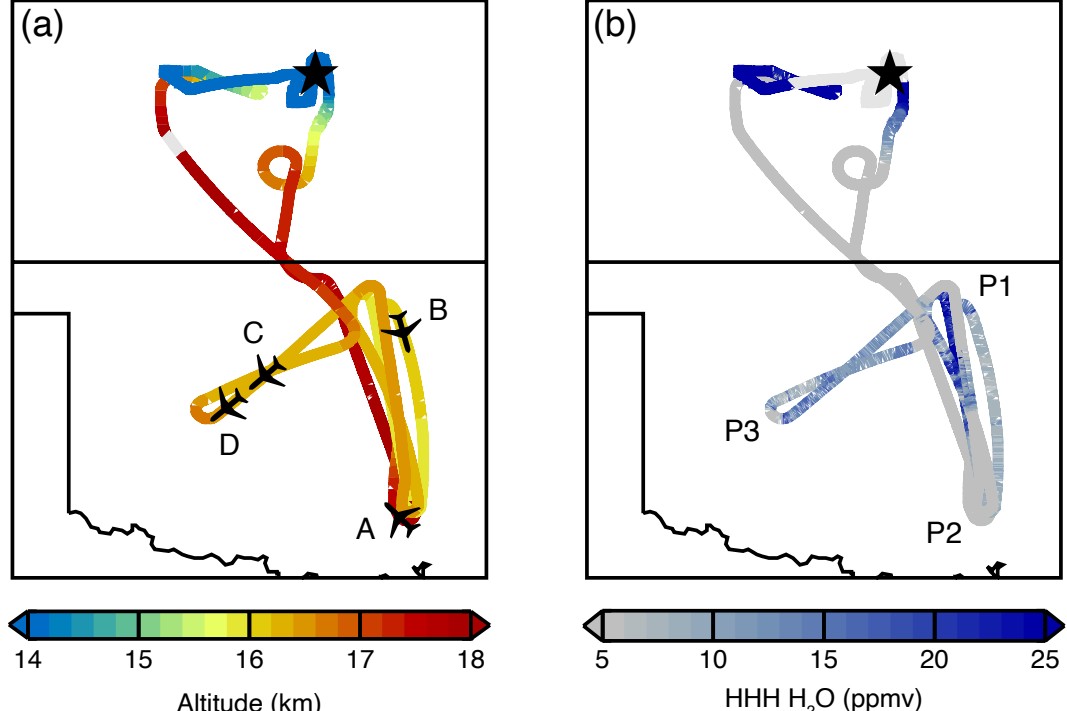

**Figure 2.** ER-2 aircraft flight track for the 31 May 2022 flight shaded by (a) altitude in km, and (b) HHH H$_2$O in ppmv. The star symbol indicates the location of Salina, KS, the base of the DCOTSS field campaign. The plane symbols labeled A, B, C, and D coincide with ER-2 camera imagery shown in Fig. 3. Point 1 (P1) is the starting point of the racetrack sampling, point 2 (P2) is the endpoint of the racetrack sampling, and point 3 (P3) was the endpoint of the AACP bisect sampling.

## 3.2 UTLS Composition Changes

Multiple segments of convectively influenced air at different altitudes were sampled during the 31 May 2022 flight. Figure 4a shows a timeseries from 00:17-02:44 UTC on 1 June of WI-ICOS total water (vapor + ice), aircraft altitude, ERA5 tropopause geopotential height, and HHH H$_2$O, with subjectively-identified convectively enhanced H$_2$O samples (relative to background concentrations at that altitude) shaded in blue. Background H$_2$O mixing ratios in the lower stratosphere range from 4 to 6 ppmv during this flight. The first two H$_2$O enhancements (convective segments 1 and 2) occur mostly below the tropopause at altitudes of 14.75 km and 15.25 km, with H$_2$O reaching near ∼30 ppmv. Convective segment 1 contains enhanced H$_2$O at two altitudes and includes the ascent between them. For both segments 1 & 2, the ER-2 is sampling near or within the tropopause-reaching anvil cloud, as revealed by total water exceeding H$_2$O periodically during these segments, indicating that ice was present in addition to vapor. Measurements where the difference between H$_2$O and total water is greater than the combined variability of the two measurements are indicative that the aircraft is within cloud material. The next two convective H$_2$O enhancements (convective segments 3 and 4) are at higher, stratospheric altitudes of ∼15.75 and ∼16.25 km (0.5-1.5 km



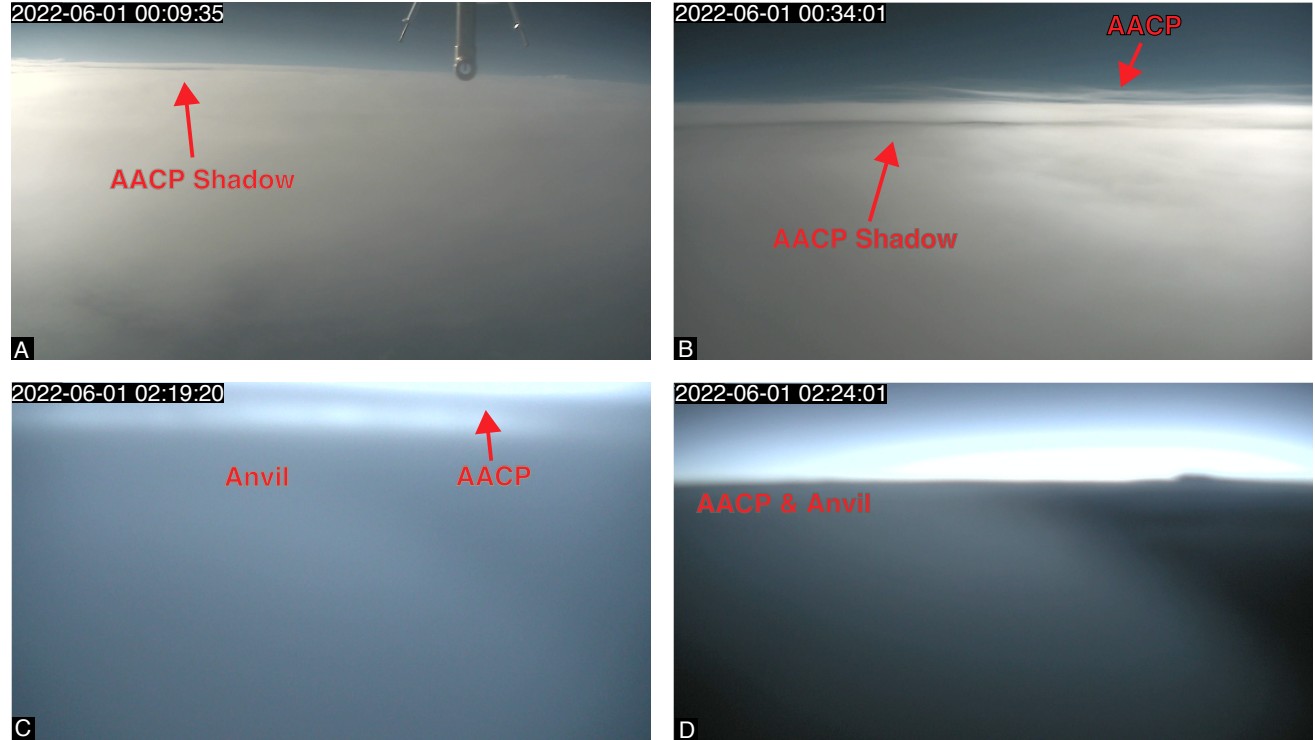

**Figure 3.** Visible imagery from the rotatable camera aboard the NASA ER-2 aircraft taken at (a) 00:09 UTC, (b) 00:34 UTC, (c) 02:19 UTC, and (d) 02:24 UTC on 01 June 2022. These images correspond to the plane symbols labeled A, B, C, and D in Fig. 2.

above the tropopause), with $H_2O$ reaching up to 20 ppmv (an approximate quadrupling of the background mixing ratio). These enhancements occur outside of cloud material, as seen by total water being approximately equivalent to $H_2O$. The final two $H_2O$ enhancements (convective segments 5 and 6) occur within the AACP bisect while the ER-2 is broadly within cloud as total water substantially exceeds $H_2O$. Elevated $H_2O$ is observed within the AACP before and during the vertical profile, with enhancements spanning 10–20 ppmv that maximize during the vertical profile.

Figure 4b is a time series including HUPCRS CO, MMS potential temperature, ROZE $O_3$, and HUPCRS $CH_4$. CO and $CH_4$, tracers of tropospheric origin, are substantially elevated above adjacent measurements within the $H_2O$ convective enhancements 1, 2, 5, and 6. These four convective segments are when the ER-2 was within cloud, skimming the anvil (convective segments 1 & 2) and bisecting the AACP on either side of the vertical profile (convective segments 5 & 6). The AACP feature will be highlighted further in subsequent analyses. It is worth noting that while $H_2O$ is enhanced at the beginning of the vertical

profile (near the end of convective segment 5), tropospheric tracers are not. This leaves segments 3 and 4, when the ER-2 is sampling outside of cloud and at stratospheric altitudes, as primarily a $H_2O$ signal. Tropospheric tracers, such as CO and $CH_4$, experience greatest convective enhancement within cloud material (convective segments 2, 5, and 6), especially within the AACP, and at closer proximity to the tropopause altitude (convective segment 1). CO reaches near 85 ppbv at an altitude of





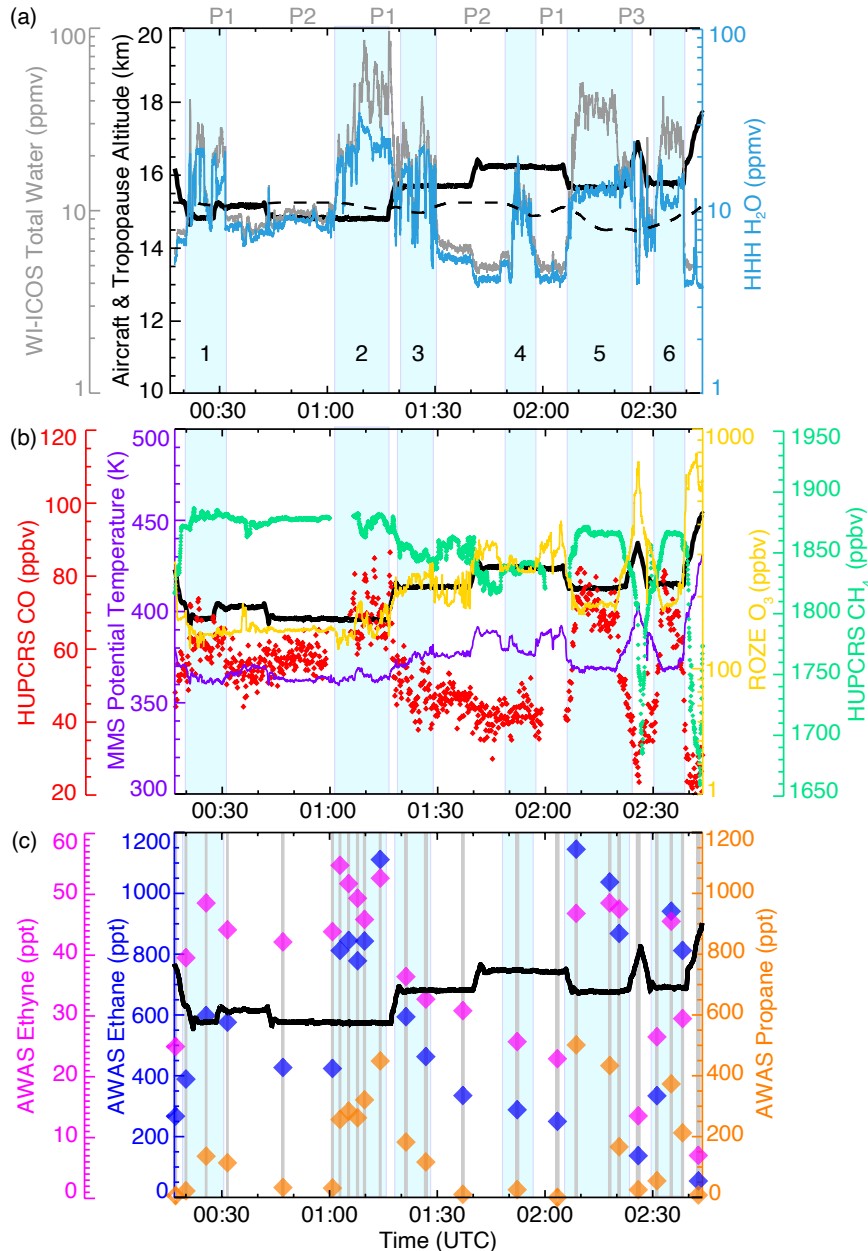

**Figure 4.** The convective sampling period (in UTC) of the DCOTSS 31 May 2022 flight shown as timeseries. All timeseries include the MMS GPS aircraft altitude (thick black line) and illustrations of the convective sampling segments (light blue shading, labeled 1–6 in timeseries (a)). The top timeseries (a) includes WI-ICOS total water (gray), ERA5 LRT altitude (dashed black line), and $H_2O$ mixing ratio (blue). Points P1, P2, and P3 from Fig. 2 are labeled (gray) atop timeseries (a). The middle timeseries (b) includes HUPCRS CO (red), MMS potential temperature (purple), ROZE $O_3$ mixing ratio (yellow), and HUPCRS $CH_4$ (green). The lower timeseries (c) includes AWAS ethyne (magenta), ethane (blue), propane (orange), and canister fill time (vertical gray bars of varying width).





14.75 km while the ER-2 is skimming the anvil beneath the tropopause, and is enhanced to ∼81 ppbv within the AACP bisect

(∼1.2 km above the tropopause). These results show that CO and CH$_4$ reach similar enhancement within the AACP compared to the anvil, even though the AACP is above the tropopause and occurs at a higher altitude and potential temperature. The maximum altitude of elevated H$_2$O for the 31 May 2022 flight occurred out of cloud material and was linked to the target storm at an altitude of ∼17 km (∼2 km above the LRT) during the final ascent to return home (at ∼02:42 UTC, which is not shaded in Fig. 4a), while the maximum altitude of elevated tropospheric tracers was 15.75 km (∼1.2 km above the LRT) and occurred

within the AACP (convective segment 6). The maximum altitude of enhanced H$_2$O relative to the ERA5 tropopause was 2.3 km and occurred out of cloud material but within the vertical profile of the AACP (near the end of convective segment 5 in Fig. 4a). Thus, it is abundantly clear from these measurements that H$_2$O enhancements reached higher absolute and tropopause-relative altitudes than tropospheric tracers, which is consistent with a common yet unexplained result from previous modeling work (Homeyer, 2015; Phoenix and Homeyer, 2021; Gordon and Homeyer, 2022). Specifically, both previous modeling work

and these observational results indicate that elevated H$_2$O can reach stratospheric altitudes at least 1 km higher than enhanced passive tracers.

Since longer-lived tropospheric tracers (CO and CH$_4$) appear to be most enhanced within cloud material, shorter-lived tracers from AWAS samples are examined as well. Figure 4c is a timeseries of AWAS ethane, ethyne, and propane. Overall, these short lived tracers exhibit similar characteristics to the CO and CH$_4$ measurements as they are most enhanced when the

ER-2 is within cloud material (predominantly convective segments 2, 5, and 6). In particular, ethane and propane are most enhanced within the AACP bisect (convective segments 5 and 6), reaching 1145 ppt and 501.4 ppt, respectively. Ethyne also exhibits enhancements throughout the 31 May 2022 flight, with a maximum of 54.6 ppt occurring within the anvil segment (convective segment 2), and additional enhancements within the AACP peaking at 48.4 ppt (convective segments 5 and 6). Enhancements in ethyne, ethane, and propane demonstrate a clear signal for stratospheric delivery of short-lived tropospheric

trace gases from oil and natural gas sources when sampling within cloud for the 31 May 2022 flight, with limited enhancement outside of cloud material. Furthermore, nearly all tropospheric gases measured during DCOTSS were enhanced within the AACP and anvil, including maximum values of multiple tropospheric gases measured during the AACP (e.g., ethane and propane shown here) rather than the anvil (e.g., ethyne, CO, and CH$_4$). This supports the distinct impact of AACP-producing storms on UTLS composition, as suggested in prior model simulations (Homeyer et al., 2017; O'Neill et al., 2021; Gordon and

Homeyer, 2022).

Given the demonstrated unique impact of the AACP on tropospheric tracer and H$_2$O enhancements in the lower stratosphere, we focus on the AACP samples and vertical profile to further examine these composition changes and gain insight into what processes could be occurring. To provide context for the detailed evaluation of AACP sampling that follows, Fig. 5 illustrates how the storm was sampled by the ER-2 and what dynamic features we would expect to encounter throughout the AACP bisect

based on insights from prior modeling work. The ER-2 bisected the AACP while traveling towards the decayed overshoot where it encountered a region with increasing turbulence, eventually resulting in a turbulence-avoidance vertical profile near the location of the decayed overshoot. The co-location of the most turbulent area with the overshoot is a common result in all modeling work focused on AACP-producing storms, owing to the establishment of a hydraulic jump that leads to gravity wave





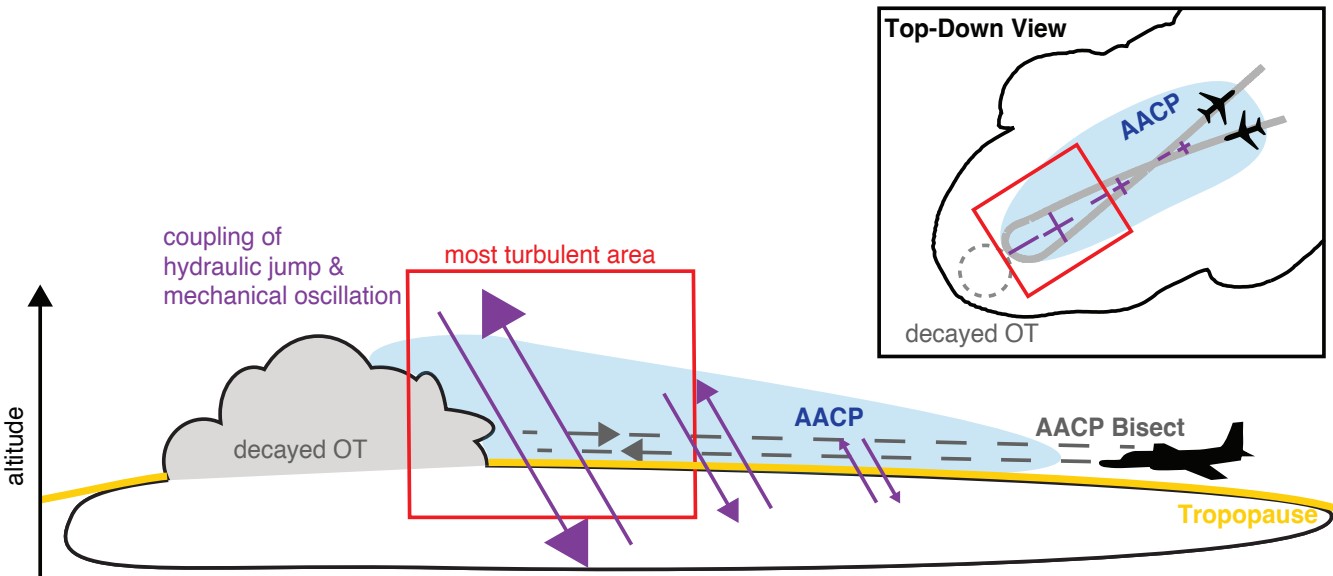

**Figure 5.** An illustration of AACP sampling during the 31 May 2022 flight. Dynamic processes we would expect to encounter (based on prior modeling work) and their feature-relative locations are indicated, including the decayed overshoot (gray), the AACP (blue), mixing associated with the coupling of the hydraulic jump and mechanical oscillation (purple), and the expected location of maximum turbulence (red). The ER-2 is shown in black, and an outline of the flight track along the AACP bisects is denoted in gray in the top-down view panel.

breaking and AACP formation (as outlined in the Introduction). This is also the location where a coupling of the hydraulic jump

and overshoot-driven mechanical oscillation lead to increased vertical mixing and enhanced downward transport of overworld air to the lowermost stratosphere (Gordon and Homeyer, 2022). Simulations show the mechanical oscillation decreases in magnitude as distance from the overshoot increases, further contributing to the area near the overshoot being the most turbulent. It is through this model-informed lens of transport and mixing processes and the known storm-relative sampling during this flight that we interpret the in situ data collected during the AACP sampling.

To examine composition changes within the AACP bisect and vertical profile components of the 31 May 2022 flight, figure 6a is a timeseries from 1:59–2:44 UTC and includes HUPCRS CO, MMS GPS Altitude, MMS Potential Temperature, ROZE $O_3$, and HHH $H_2O$. The AACP bisect begins at ~02:07 UTC and continues until the turbulence-avoidance vertical profile occurs from 02:23–02:28 UTC, and resumes thereafter until 2:39 UTC when the ER-2 begins to ascend to maximum altitude and return to Salina. Tropospheric tracers, such as CO, are elevated within both passes through the AACP, with an exception

near the ends of the horizontal segments bounding the vertical profile. Potential temperature and $O_3$ begin to increase at 02:20 UTC, shortly before the vertical profile begins and when altitude is constant. The increase in stratospheric tracers and decrease in tropospheric tracers (except for $H_2O$) at constant altitude suggests a reduction in the amount of tropospheric air transport to the stratosphere (i.e., a $H_2O$-dominant convective signature). This apparent "stratospheric signal" at constant altitude (annotated in Fig. 6a) implies a change in the physical/dynamic process leading to convectively influenced air above the



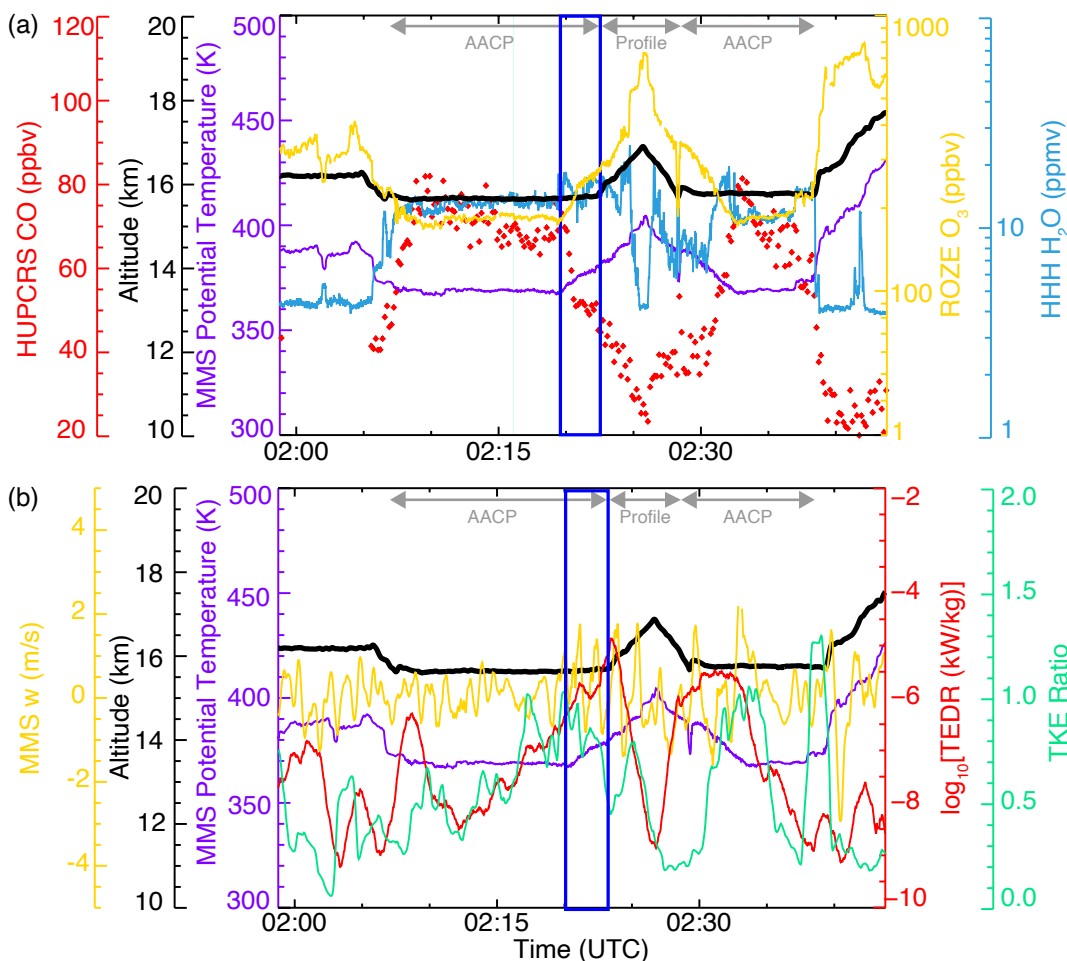

**Figure 6.** AACP sampling during the 31 May 2022 flight (in UTC) shown as timeseries. Both timeseries include the MMS GPS aircraft altitude (thick black line) and potential temperature (purple line). The top timeseries (a) uniquely includes HUPCRS CO (red), ROZE $O_3$ mixing ratio (yellow), and HHH $H_2O$ mixing ratio (blue). The lower timeseries (b) uniquely includes MMS measurements of the $\log_{10}$ of the Turbulent Eddy Dissipation Rate (TEDR; red), the vertical-to-horizontal TKE ratio (green), and the vertical wind speed (m/s; yellow). The dark blue box in each timeseries denotes the time period of the 'stratospheric signal' near the end of the first AACP bisect, and gray horizontal lines with arrows at each end denote times when the ER-2 is within the AACP and in the turbulence-avoidance vertical profile.



tropopause. A change in controlling processes is also likely present within the second AACP bisect, as potential temperature and $O_3$ are elevated and then decrease while at constant altitude. Note that inspection of ERA5 output suggests there are no alternative large-scale explanations for an increase in potential temperature from a pre-existing isentropic gradient in this region. Therefore, the stratospheric signal, evident as a unique signature in trace species concentrations distinct from those in the broader AACP feature, suggests different mixing processes are occurring near the end of the first AACP bisect and the

beginning of the second AACP bisect (02:28–02:33 UTC) closest to the decayed overshoot.

To identify what processes are primarily driving the stratospheric signal (corresponding with the time that substantial turbulence was encountered, resulting in the eventual ascent of the ER-2 out of the AACP), we examine the turbulent energy dissipation rate (TEDR), turbulence kinetic energy (TKE), and vertical wind speed measured by MMS (w; Fig. 6b). TEDR combines both vertical and horizontal components of turbulent motion in one metric, and was smoothed with a 90-second

centered average to show turbulent bursts throughout the flight. Horizontal and vertical TKE was calculated using MMS wind speeds after application of a high-pass filter (removing power at time scales greater than 4 seconds to isolate turbulent motions) followed by calculating the 20-second variance of these filtered winds. A 90-second centered average was then applied to the horizontal and vertical TKE components before calculating the ratio (vertical:horizontal) shown in Fig. 6b. These time intervals were chosen to best capture the spatial and temporal differences throughout the flight. The absolute vertical winds are also

shown to provide additional context, which were calculated using a low pass filter (removing power at time scales less than 4 seconds, thus emphasizing the broader scales of motion including those at the scale of gravity waves and the hydraulic jump) and applying a 20-second average to be consistent with the 20-second variance used for TKE calculations.

Converting displayed $\log_{10}$(TEDR) values to TEDR dissipation rates throughout this portion of the flight shows an increase in turbulent motion within the stratospheric signal prior to the vertical profile (near $10^{-2}$ W/kg at 02:20 UTC), and within

the AACP following the vertical profile (near $3.2*10^{-3}$ W/kg at 02:31 UTC). This suggests that peak turbulence is occurring within the stratospheric signal and near the decayed overshoot, coinciding with the turbulence-avoidance vertical profile. This is consistent with what is expected based on Fig. 5, where the most turbulent region within the AACP is located near the overshoot. Examining the TKE ratio within these areas of elevated TEDR provides additional insight towards the contributions of vertical and horizontal mixing to the increasing turbulence. For the majority of the 31 May 2022 flight, horizontal TKE

exceeds vertical TKE, as indicated by the TKE ratio being consistently below a value of 1. However, there is a rapid increase in TKE ratio to a value near 1 shortly prior to and within the stratospheric signal during the first (02:17–02:23 UTC) and second (02:31–02:35 UTC) AACP bisects. This suggests that increased vertical mixing is associated with the emergence of the stratospheric signal. It is worth noting that the TKE ratio also increases to a value near 1 (suggesting increased vertical mixing) within elevated TEDR at a flight time of approximately 02:31 UTC. This coincides with sampling the AACP cloud boundary

following descent from the short vertical profile, which resides within the most turbulent area of the AACP near the decayed overshoot (as seen in Fig. 5). Similar trace gas changes, including decreasing $O_3$ with increasing $H_2O$ and CO, are seen near these vertical TKE peaks in Fig. 6a. Vertical winds (w) provide additional context towards the contributions of vertical mixing with turbulent regions. Throughout the majority of the sampling period shown in Fig. 6b, w oscillates between -0.5 and 0.5 m/s. However, the oscillation increases in magnitude throughout the first AACP bisect, and reaches a range of near -1 to 1 m/s



at the onset of the stratospheric signal (when the ER-2 is near the decayed overshoot). Similarly, w shows a strong oscillation within the second AACP bisect following the descent from the vertical profile ranging from -1.5 to 2 m/s, when the ER-2 is near the decayed overshoot. A stronger oscillation and vertical mixing near the decayed overshoot is in agreement with the key dynamic mechanisms illustrated in Fig. 5, where vertical mixing associated with the coupling of the hydraulic jump from the AACP and overshoot-induced mechanical oscillation is strongest near the overshoot and decreases in strength as distance from the overshoot increases. Given this close coupling between prior simulations and the DCOTSS measurements, we therefore believe that the stratospheric signal represents the downward mixing of stratospheric air from higher altitudes in the vicinity of the coupled hydraulic jump and mechanical oscillation.

Next we examine relationships between $O_3$ and $H_2O$ during the AACP bisect and in the segment with the stratospheric signal where the $O_3$ mixing ratio increases. $O_3$-$H_2O$ correlations for the AACP bisect and profile components of the 31 May 2022 flight are shown in Fig. 7a. Strong linear correlations were identified via the Pearson correlation coefficient, which was calculated over an observation-centered time window of $\pm 10$ seconds. A strong positive correlation is considered to be $\geq$ 0.5 and, similarly, a strong negative correlation is $\leq -0.5$. There are many strong negative and positive $O_3$-$H_2O$ correlations throughout the AACP bisect. Generally, with overshooting convection it is expected that air of high $H_2O$, cloud ice, and low $O_3$ composition will mix with stratospheric air of high $O_3$ and low $H_2O$ composition, resulting in a negative $O_3$-$H_2O$ correlation (Fig. 8). Therefore, it is not surprising to see the frequency of negative $O_3$-$H_2O$ correlations associated with STE, but the occurrence and frequency of positive $O_3$-$H_2O$ correlations was unexpected. There are multiple positive correlations coinciding with the AACP bisect and at the onset of stratospheric signal. To better understand the potential factors leading to positive $O_3$-$H_2O$ correlation, we further examine multiple tracers and meteorological variables.

Figure 7b shows a time series for the AACP bisect and profile components of the 31 May 2022 flight including MMS temperature, the saturation mixing ratio (SMR), HHH $H_2O$, and WI-ICOS total water (vapor + ice). During the first AACP bisect (flight time of 02:07-02:22 UTC), temperature broadly decreases to $\sim$199 K, followed by a sharp increase to $\sim$206 K during the stratospheric signal (beginning at 02:20 UTC), consistent with sampling a warmer stratospheric environment. Total water also drops at the onset of the stratospheric signal from 30-40 ppmv to 10-20 ppmv, while $H_2O$ remains elevated between 10 and 20 ppmv. Such changes in total water and $H_2O$ at this time suggest that mixing of AACP air with the warmer stratospheric environment is significant, which is in agreement with prior analyses and with the broad occurrences of negative $O_3$-$H_2O$ correlations at the beginning and end of this time segment. In contrast, when the temperature is low throughout the remainder of the first AACP bisect, $H_2O$ is nearly equal to the SMR, demonstrating that the air is saturated and $H_2O$ within the convectively lofted air is constrained to the SMR. Notably, the $O_3$-$H_2O$ correlations observed during this saturation-limited period of sampling are almost entirely positive. When the temperature increases at the onset of the stratospheric signal, the SMR exceeds $H_2O$ and the air becomes subsaturated while the $O_3$-$H_2O$ correlation sign flips to negative. When the temperature rapidly decreases following the vertical profile and near the start of the second AACP bisect, total water rapidly increases to values exceeding $H_2O$ and the AACP air is once again saturated. Similar patterns in the $O_3$-$H_2O$ correlations are also observed within the second AACP bisect. Given these relationships, a plausible explanation for the occurrence of positive $O_3$-$H_2O$ correlations is that $H_2O$ within the overshoot and AACP air is saturation-limited due to the low temperature within



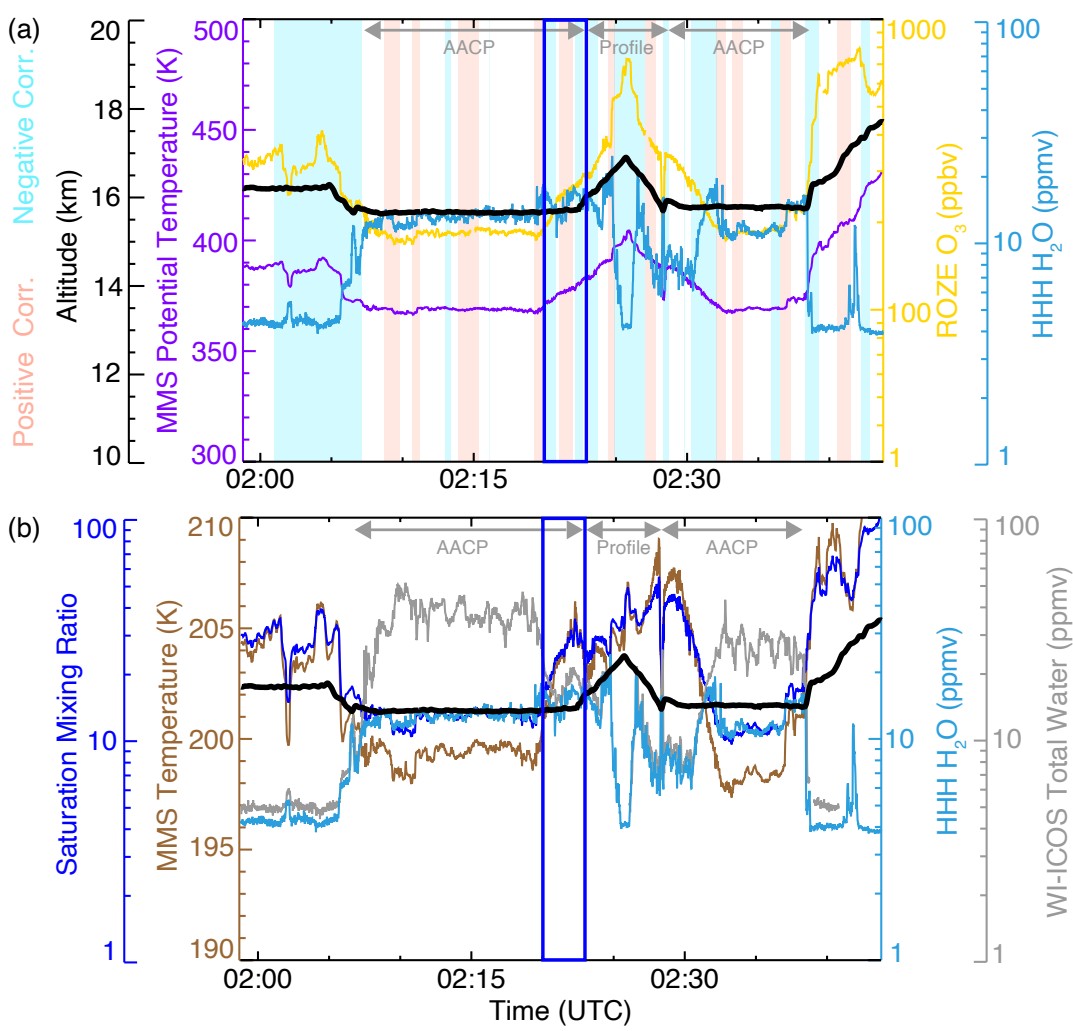

**Figure 7.** As in Fig. 6, but with top timeseries (a) of MMS GPS altitude (black), MMS potential temperature (purple), ROZE O$_3$ mixing ratio (yellow), HHH H$_2$O mixing ratio (blue), positive O$_3$-H$_2$O correlations (blue shading), and negative O$_3$-H$_2$O correlations (pink shading). The lower timeseries (b) includes MMS GPS altitude (black), MMS saturation mixing ratio (dark blue), MMS temperature (brown), H$_2$O mixing ratio (light blue), and WI-ICOS total water (gray).



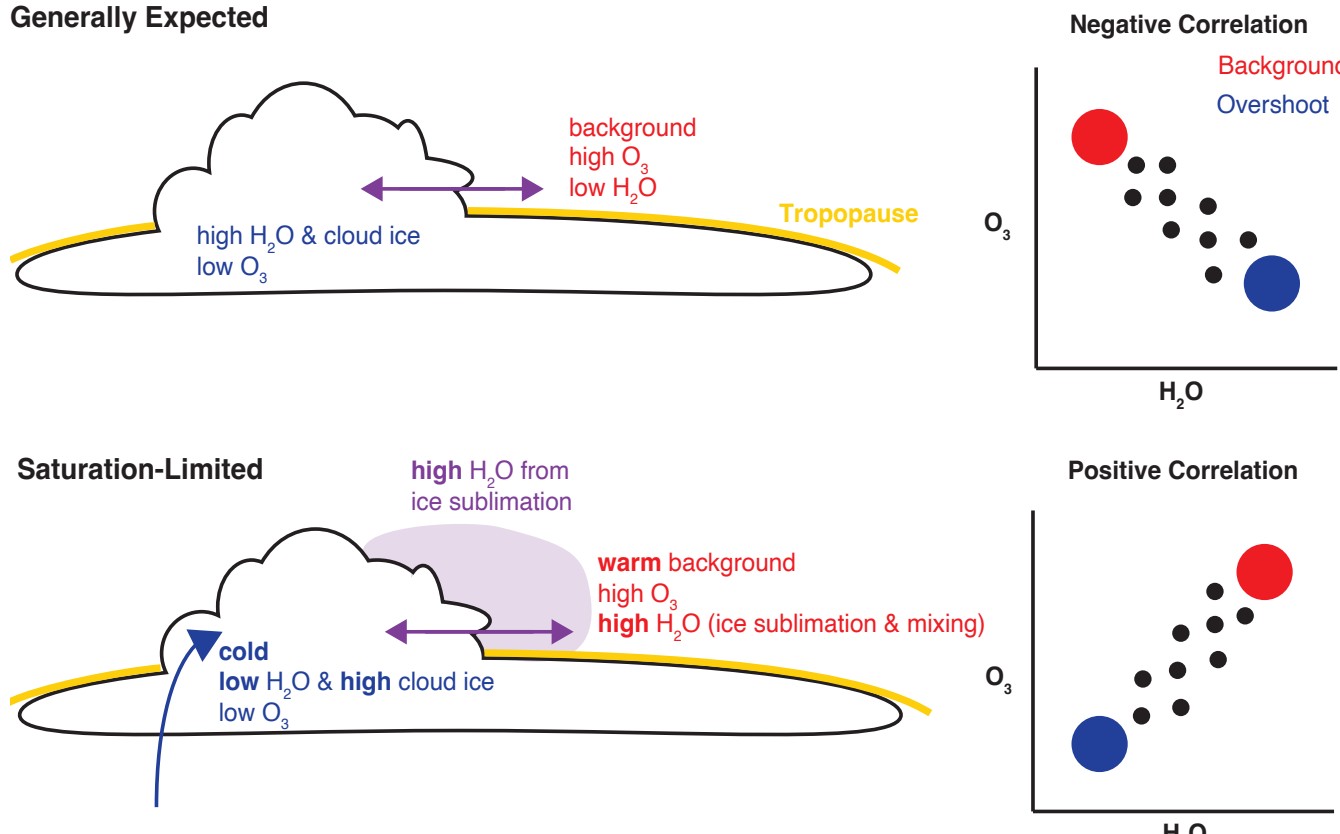

**Figure 8.** Illustration of the process that is generally expected for an overshoot which results in a negative $O_3$-$H_2O$ correlation (top panels), and a saturation-limited overshoot and hypothesized process resulting in a positive $O_3$-$H_2O$ correlation (bottom panels).

the overshoot. In the absence of a saturation-limited condition at low temperatures, there is significantly more $H_2O$ in the overshoot than the stratospheric environment. However, with these particular overshoots, the temperature decreases to a point where the SMR is less than the maximum possible $H_2O$ concentration in the environmental stratospheric air, resulting in less water in the vapor phase and more in the ice phase within the overshoot. To account for this $H_2O$ imbalance, the cloud ice along the periphery could preferentially sublimate into the warm stratospheric environment (rather than remaining in the overshoot),

while subsequent mixing between this hydrated stratosphere air and the relatively dry overshoot leads to a positive $O_3$-$H_2O$ correlation, as illustrated in Figure 8.

While the previous analyses have been focused on understanding the dynamic and physical processes constraining stratospheric hydration and STE, one question that remains is whether the 31 May 2022 AACP observations differ from the typical range of lower stratosphere composition impacts from overshooting convection (AACP-producing or otherwise). Therefore, to

assess the uniqueness of these active convection observations, measurements of $H_2O$ and CO from the 31 May 2022 flight are compared to samples of overshooting convection from all other DCOTSS research flights. Figure 9 shows the density of con-



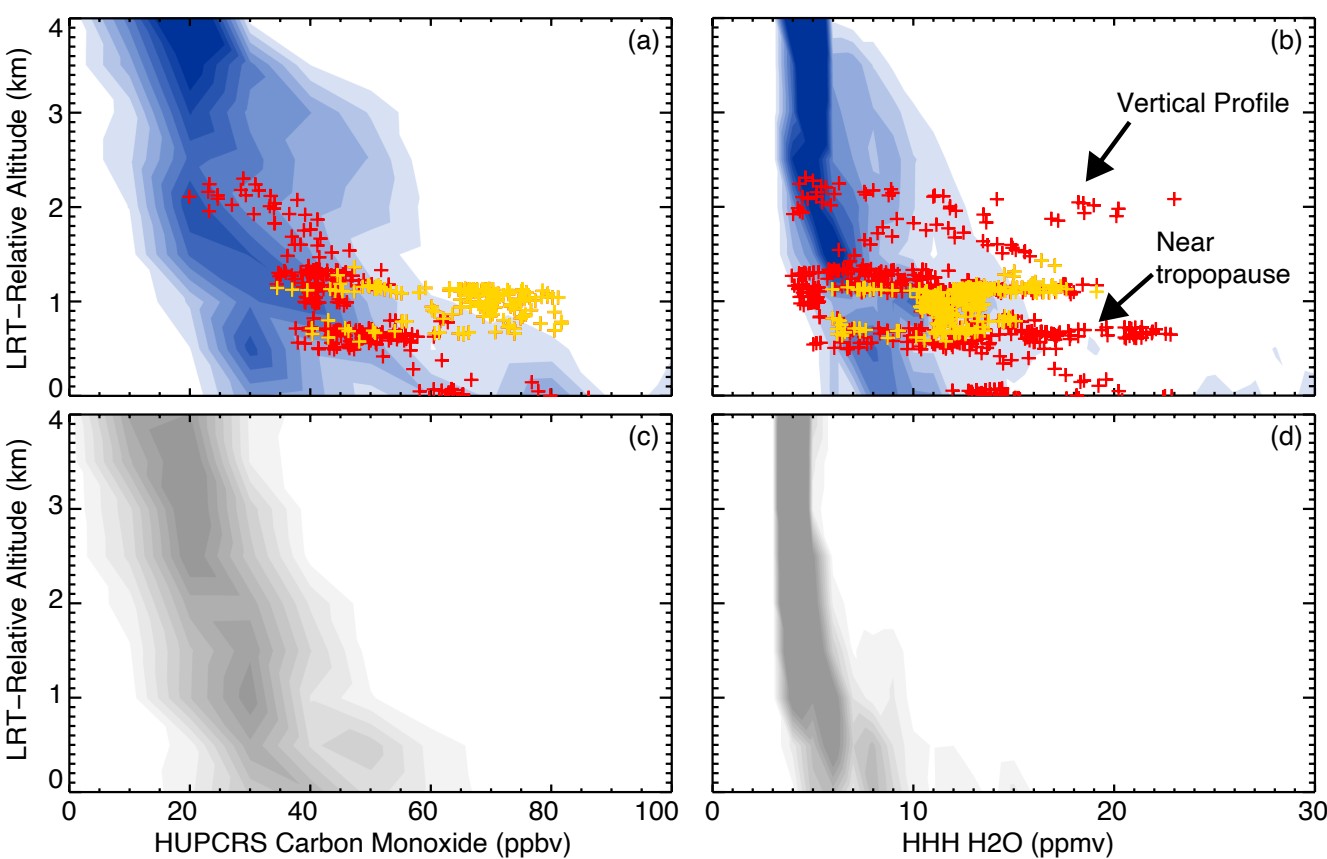

**Figure 9.** Density of CO (left) and $H_2O$ (right) in LRT-relative altitude of convective segments (blue; a and b) and background segments (gray; c and d) from all DCOTSS 2021 and 2022 flights except for the 31 May 2022 flight. Convective sampling from 31 May 2022 flight (red) and AACP sampling from 31 May 2022 flight (yellow) is shown atop the respective convective sampling densities. Density is determined using a 10.0 ppbv x 0.5 km bin resolution for CO and a 1.0 ppmv x 0.5 km bin resolution for $H_2O$. For the 31 May 2022 convective samples, every 5th $H_2O$ measurement is shown, to be more consistent with the frequency of CO. Measurements obtained when skimming the anvil near the tropopause and during the vertical profile are annotated.




vective observations as a function of trace gas concentration and ERA5 LRT-relative altitude, with the convective observations (red) and AACP observations (yellow) from the 31 May 2022 flight overlaid. For additional context, the density of background (i.e., non-convective) observations from all DCOTSS flights except for the 31 May 2022 flight are shown in additional panels.

Background DCOTSS observations of CO range from 15-70 ppbv near the tropopause and 5-50 ppbv throughout much of the stratosphere. Convective samples from remaining DCOTSS flights span approximately 25-90 ppbv near the tropopause and 10-60 ppbv throughout much of the lower stratosphere (up to ∼3 km above the tropopause). Most of the enhanced CO (>40 ppbv) in convective samples at an LRT-relative altitude of 2-3 km originate from the 08 June 2022 and 10 June 2022 flights that targeted recent/active convection (not shown). Measurements from the 31 May 2022 active convection flight show

an above-average mixing ratio of CO in overshoot material, and exceptional values within the AACP at a LRT-relative altitude of 0.5-1.2 km. The majority of $H_2O$ convective sampling for all DCOTSS flights ranges from 3-15 ppmv, while the density of background observations is 3-5 ppmv throughout much of the stratosphere, especially at LRT-relative altitudes >1 km. The 31 May 2022 active convection flight sampling of $H_2O$ is largely consistent with convective measurements from other DCOTSS flights, with greater extremes than those commonly measured at altitudes near the tropopause and >1.5 km above

the tropopause. The most prominent example of exceptional $H_2O$ observed during the 31 May 2022 flight is near a LRT-relative altitude of ∼2.1 km where $H_2O$ reaches ∼23 ppmv. This $H_2O$ enhancement coincides with the stratospheric signal at constant altitude and subsequent vertical profile as the ER-2 emerged from the first AACP bisect, which was primarily a $H_2O$ convective signal (as tropospheric tracers were not strongly enhanced). $H_2O$ sampled within the AACP bisects is near or slightly above the typical range of $H_2O$ mixing ratios for all DCOTSS convective sampling at the LRT-relative altitudes spanned by the AACP.

As demonstrated in Fig. 8, $H_2O$ concentrations in the AACP are limited by the SMR set by the low temperatures in the AACP air, making these enhancements less exceptional than those seen with CO compared to remaining DCOTSS flights.

## 4 Conclusions

This study analyzed observations from the DCOTSS 31 May 2022 flight obtained within convective outflow and an actively-generated AACP from a storm over Oklahoma. UTLS composition changes from the storm were evaluated and discussed in

comparison with results from recent model simulations. Multiple segments of convectively influenced air with pronounced changes in stratospheric hydration and air mass composition were sampled throughout the flight. Consistent with previous modeling studies, observations showed that water vapor enhancements reached higher altitudes than passive tropospheric tracer enhancements (Figs. 4 & 6a). Convective enhancements of short-lived tracers including CO, ethyne, ethane, and propane were found to be most enhanced when sampling within cloud material, specifically when bisecting the storm anvil and AACP (Figs.

4 & 6a). This suggests that AACP-producing storms have a unique impact on UTLS composition, especially with respect to troposphere-to-stratosphere transport of tropospheric air. In addition to enhanced tropospheric tracers within the AACP, there was an apparent "stratospheric signal" at constant altitude near the end of the first AACP bisect and beginning of the second AACP bisect, where $O_3$ and potential temperature increased while CO, $CH_4$ and total water decreased and $H_2O$ remained elevated (or vice versa). This stratospheric signal seems to be associated with enhanced vertical mixing, based on both composition



changes and turbulence analyses (Fig. 6b). Observations of vertical mixing and downward transport of deeper stratospheric air near the AACP edge (in close proximity to the once-active overshooting top) supports the novel finding in recent modeling work (Gordon and Homeyer, 2022). $O_3$-$H_2O$ correlations were examined throughout the flight, and multiple fine-scale positive and negative correlations were observed. There are notable positive $O_3$-$H_2O$ correlations throughout the AACP bisects that appear to be the result of preferential sublimation of cloud ice into the warm stratospheric environment and subsequent mixing between overshoot/AACP and hydrated stratosphere air (illustrated in Fig. 8). Tropospheric tracer enhancements including CO and $H_2O$ observed from the 31 May 2022 flight are unique compared to remaining DCOTSS flights. While $H_2O$ enhancements observed during the AACP bisect were not extraordinary in comparison to remaining observations during the flight due to the aforementioned saturation-limited condition, CO measurements within the AACP are exceptional when compared to remaining DCOTSS observations (Fig. 9). These comparisons support the impactful nature of AACPs on air mass composition within the UTLS inferred from model simulations, in addition to their increasingly-appreciated impact on stratospheric hydration.

The observational results in this study emphasize that determining how often overshooting storms produce AACPs is important for assessing both $H_2O$ and non-$H_2O$ composition impacts to the UTLS. While some insight into stratospheric hydration can be gained from existing overshoot climatologies (Homeyer and Bowman, 2021; Cooney et al., 2018; Solomon et al., 2016; Liu and Liu, 2016; Liu et al., 2020), an AACP climatology is desperately needed to provide sufficient knowledge towards $H_2O$ and non-$H_2O$ UTLS composition change. Machine learning efforts are underway that may result in an objective AACP climatology.

While this study is the first to analyze targeted observations of an AACP, it is important to note that this was the only DCOTSS flight to do so. Thus, these observations from an AACP-producing storm are representative of one storm within one environment. Observed UTLS composition changes may vary substantially in other scenarios based on the known sensitivities of STE to the stratospheric environment. The DCOTSS observations from this flight also have a limited spatial extent, with observations not spanning the entirety of the storm throughout its lifetime. It is also worth noting that it is possible some of the overshoot-influenced air measured could have been sourced from the overshooting storm to the southwest of the target storm. However, the premise of this analysis would not change as the analyzed observations are from active overshooting convection and an AACP regardless of the responsible storm cell. Future modeling work could also simulate this event to supplement the observational analysis conducted here, providing additional perspective on the UTLS composition changes, sources, and constraining mechanisms.

Observations from the 31 May 2022 flight showed that $H_2O$ within the overshoot and AACP was saturation-limited due to the constraint of the low temperature, and that stratospheric hydration was occurring via sublimation and subsequent mixing of cool, high-ice cloud material into the warm stratospheric environment. Thus, exploring a saturation-limited perspective on $H_2O$ delivery from overshooting convection is important for future work. Specifically, it is important to determine how frequently a $H_2O$-limiting temperature constraint occurs and its impact on stratospheric hydration. It is currently unclear whether the temperature constraint is driven by environmental characteristics, or if the storms themselves can facilitate it. Exploring the microphysics occurring in overshoots and AACPs could provide increased understanding of these processes. The saturation-limited condition was also associated with a uniquely positive correlation between $O_3$ and $H_2O$. Ongoing efforts evaluating



$O_3$-$H_2O$ correlations from all DCOTSS flights may provide indirect evidence for the importance of this temperature constraint on a broader scale.

*Data availability.*    All DCOTSS aircraft data NASA (2023a), and satellite & radar data NASA (2023b) analyzed here are available from the NASA Atmospheric Sciences Data Center: https://asdc.larc.nasa.gov/project/DCOTSS.

*Author contributions.*    A. Gordon and C. Homeyer designed and executed the analysis, with substantial design contributions from J. Smith,
R. Ueyama, and J. Dean-Day. All instrument data presented here were obtained and rigorously analyzed and carefully processed to ensure QA/QC by co-authors J. Smith, R. Ueyama, J. Dean-Day, E. Atlas, K. Smith, J. V. Pittman, D. S. Sayres, D. M. Wilmouth, A. Pandey, J. St. Clair, T. Hanisco, J. Hare, R. Hannun, S. Wofsy, B. Daube, and S. Donnelly. A. Gordon drafted the manuscript and all co-authors provided editorial comments and corrections.

*Competing interests.*    The authors declare that they have no conflict of interest.

*Acknowledgements.*    The authors thank the entire DCOTSS team for being instrumental in planning and executing the successful flights and measurements discussed here. This work and all authors were supported by the National Aeronautics and Space Administration (NASA) under Earth Venture Suborbital-3 program awards for DCOTSS (NASA Awards 80NSSC19K0347, 80NSSC19K1473, 80NSSC19K0326, and 80NSSC19K0340).



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
