# Peer review of "Detailed Examination of Upper Troposphere Lower Stratosphere Composition Change from DCOTSS Airborne Observations of Active Convection on 31 May 2022"

_EGUsphere, 2023_

## Author Comment (AC1)

**Response to Reviewers for EGUSPHERE Paper #2023-2603**

We thank the reviewers for their insightful comments, which helped to improve the manuscript. Detailed responses to each Reviewer's comments can be found in the following pages (reviewer comments in **black**, our responses in **blue**).

**Reviewer #1:**

This paper presents measurements from the recent DCOTTS aircraft campaign over the US, which aimed at sampling deep convective storms and investigating their impact on the UTLS. The focus is on one particular flight (31 May 2022) and on the convective enhancement of several tropospheric tracers and water vapor. It is shown that within the convectively influenced air the water vapor mixing ratios are significantly enhanced, up to about 2km above the local tropopause. Special emphasis is laid on the measurements within above anvil cirrus plumes (AACP), which were observed during that flight. Water vapor versus ozone correlations together with ice enhancements show that in the AACP air masses the convective moistening is saturation-limited, and that vertical mixing with the ambient air frequently occurs.

I find this an interesting study on an extreme case of convective impact on the UTLS region which, together with the presentation of new observations, clearly falls within the scope of the journal. The paper is well written and the presentation is clear. I have a few minor and specific comments below which will hopefully further improve the paper and which needs to be addressed before publication.

Minor comments:

1. Correlation analysis (L350ff and Fig. 8):

I'm not entirely convinced how meaningful the relation between the calculated correlation coefficient and the dehydration/moistening process is. For instance, how to explain the frequent changes of correlation characteristics over short time periods? If a moist plume is characterized by both positive and negative correlations (e.g. the discussed about 23ppmv enhancement in the ascending leg of the profile around 02:25 UTC, or in the second AACP leg around 02:32 UTC) how to interpret that - or, in other words, by which process is it caused? Also, some of the positive correlations occur during flight times when the air is largely subsaturated (e.g. at the end of the first AACP leg around 02:22 UTC or during the descending profile leg) - so how can the related moistening process be interpreted as being "saturation-limited"? Could it be that additional noise in the timeseries blurs the physical signal in the correlation to some degree? Or could these issues be related to the "subjectively identified convection" and could be resolved with reconsidering the used criterion (see my specific comment below)? It would be good to add some sensitivity analysis  to show that  the calculated correlation coefficients are indeed physically meaningful, and/or to discuss the uncertainties and limitations.

The positive correlations in a subsaturated environment are not a result of the subjective identification of several convective segments - see the response to that specific comment below for more details on this. Otherwise, we have added a few sentences to aid in the meaning, interpretation, and reliability of this correlation analysis within the paragraph discussing Figure 7a. While not shown, the sensitivity to time window used is small up to times as large as a minute. Bias in assessed correlations will increase with longer times since measuring correlations that result from pre-existing large-scale gradients in the trace gases becomes more likely. Correlations during profile segments are not reliable for similar reasons - the pre-existing vertical gradient in trace gases (especially ozone and water vapor) is large, such that prevailing correlations will exist regardless of the unique impacts resulting from physical or dynamic processes.

2. Global impact (abstract, conclusions):

The relevance of convection and in particular AACP-producing storms for the global scale is to some degree discussed in the conclusions section. However, I find that this discussion could be enhanced. In this context, also the sentence in the abstract "Such UTLS composition changes alter the radiation budget ..." appears to me as if the impact of these convective events on global scale would be given. My personal impression here is that from a global perspective the occurrence frequency of these types of deep storms is not very high and the amount of tracers and water vapor which could be released into the stratosphere will be rather limited. Sure, a thorough investigation of this impact is an independent project. But are there any published studies which aim at upscaling the impact of such storms to global scale, and what impact could be expected? At least, the potential impact on the global scale could be discussed more critically here.

Thank you for pointing this out. We believe that discussing prior work on the topic is best placed within the Introduction. Thus, we have added the following to lines 64-71 of the revised manuscript:

*"The contribution to the global lower stratosphere $H_2O$ budget from overshooting convection is currently estimated to range from 10%-15%, based on several studies using output from convection-resolving model simulations and observations of storms with a trajectory model driven by large-scale winds  (Dauhut & Hohenegger 2022, Dessler & Sherwood 2004, Tinney & Homeyer 2021, Ueyama et al 2023). This is likely an underestimation, as most existing efforts to quantify the lower stratosphere $H_2O$ budget only consider effects from tropical convection. However, there has been increasing attention towards midlatitude convection contributions to the lower stratosphere $H_2O$ budget, due to its observed frequency and depth (Khaykin et al 2022, Liu & Liu 2016, Liu et al 2020, Werner et al 2020)."*

Specific comments:

L36: I'm somewhat surprised about this high fraction of overshooting convection reaching the overworld (50%). Can you add an appropriate reference for that?

> The appropriate reference of Cooney et al 2018 has been added to line 39.

L172: I'm also surprised that using 37 pressure level data instead of the original 137 level data has no effect on tropopause detection and the presented results. Could you explain further why, and could you state the pressure levels among the 37 which are in the UTLS region? Given the importance of the estimated tropopause altitude for the presented results it could be worth explicitly showing this insensitivity (e.g. in an appendix figure).

> As we state in the paper, the refinement is minor (not zero, but small). The vertical grid spacing in the UTLS for the 37 pressure level grid is ~1 km, but the underlying temperature lapse-rate change which facilitates tropopause identification is well-recovered by cubic spline interpolation (as demonstrated in the studies cited). Specifically, we have found that differences between tropopauses based on cubic-spline interpolated 37 pressure level volumes and those computed on the 137-level native model grid are typically less than ±500 m (see scatterplot below; this is a standard and tolerable level of uncertainty for the analyses presented). There are some larger differences near the tropopause breaks where the tropopause altitude is very uncertain in any gridded dataset, but the vast majority of points fall within the ±500 m lines and the r.m.s. error for this comparison is ~500 m. We have reworked some of this very short Section 2.3 to better capture these details and justify our choices, with a few additional citations.

[Figure]

L226: What does "subjectively-identified convectively enhanced" here means? On what criteria was this identification based? Please clarify and describe in the manuscript. Could this subjective identification cause issues for the correlation analysis (see my minor comment above)?

      This text has been revised to "subjectively-identified convective enhancements in…". It means we drew boxes in the figure to guide the eye of the reader, based upon water vapor observations that greatly exceeded background mixing ratios for several minutes (as defined in the text here, but we've clarified the language a bit - lines 245-246 of the revision). Thus, the subjective identification is independent of the correlation analysis and would not impact it, as the correlation analysis focuses on $H_2O$ and $O_3$ relationships, temperature, and saturation from the 1-second data, rather than subjectively-identified convection segments.

L283: Please explain here in more detail how the "hydraulic jump" is characterized.

      The following text has been added to lines 53-54 of the revised manuscript, where 'hydraulic jump' is first mentioned in the Introduction (see Reviewer 2 comment): *"This hydraulic jump occurs near the tropopause and downstream of the overshooting top, where the overshooting top acts as a topographic obstacle."*

L299: It would be helpful to explain here more clearly what exactly is meant by "change in the physical/dynamic process"? I guess you refer here already to the onset of turbulent mixing, as discussed further below - but this could be mentioned here already.

      We have decided to leave the text as is, since this is an intentionally vague transition in the discussion to motivate the detailed analysis that follows.

L396: When looking at Fig. 7, the 23 ppmv moist plume seems to be entirely within the ascending leg of the profile. Please clarify the text here.

      Yes, the 23 ppmv maximum occurs within the profile. We have revised this text for clarity (lines 423-424 of the revision).

Citation: https://doi.org/10.5194/egusphere-2023-2603-RC1

**Reviewer #2:**

General comments

The submitted manuscript presents and analyses detailed observations obtained from one particular flight of an aircraft campaign, during which, quite remarkably, measurements of atmospheric composition were made along a novel bisect of an above-anvil cirrus plume located well above the tropopause. This detailed study clearly adds to our knowledge of UTLS processes and composition and transport across the tropopause.

The manuscript is very carefully and clearly written, and firmly within the scope of ACP and in my opinion is very suitable for publication. I have only quite minor comments.

It is not always easy to present the wealth of data collected during a well-instrumented flight such as this in a way that is clear and insightful to the reader, but the authors have tried very hard, in figures such as 4 ,6 and 7 with their choice of plots and helpful annotations. (Some reviewers might object to figure 8 as being too pedagogical but personally I liked it.)

The reasoning seemed logical in all places and based on the data collected, although of course, as discussed by the authors in the conclusion, it is not clear how far the findings based on one particular event can be extrapolated or be considered representative of the general process. Hopefully, further campaigns will be undertaken to see how reproducible the features found here are at different times and places.

Specific comments

Although the introduction was nicely done, I felt the range of references was perhaps slightly narrow with too much of a focus on the research of the co-authors Some examples of recent work being carried out by other groups include:

Nugent, J. M., & Bretherton, C. S. (2023). Tropical convection overshoots the cold point tropopause nearly as often over warm oceans as over land. GeophysicalResearch Letters, 50, e2023GL105083. https://doi.org/10.1029/2023GL105083

Spang, R., Müller, R., and Rap, A.: Radiative effect of thin cirrus clouds in the extratropical lowermost stratosphere and tropopause region, Atmos. Chem. Phys., 24, 1213–1230, https://doi.org/10.5194/acp-24-1213-2024, 2024.

Clapp, C. E., Smith, J. B., Bedka, K. M., and Anderson, J. G.: Distribution of cross-tropopause convection within the Asian monsoon region from May through October 2017, Atmos. Chem. Phys., 23, 3279–3298, https://doi.org/10.5194/acp-23-3279-2023, 2023.

Khaykin, S. M., Moyer, E., Krämer, M., Clouser, B., Bucci, S., Legras, B., Lykov, A., Afchine, A., Cairo, F., Formanyuk, I., Mitev, V., Matthey, R., Rolf, C., Singer, C. E., Spelten, N., Volkov, V., Yushkov, V., and Stroh, F.: Persistence of moist plumes from overshooting convection in the Asian monsoon anticyclone, Atmos. Chem. Phys., 22, 3169–3189, https://doi.org/10.5194/acp-22-3169-2022, 2022.

(To be clear, I am only offering these as examples, by no means insisting that you cite them).

Thanks for this suggestion - additional references have been added to the Introduction. These include Frey et al 2015 and Khaykin et al 2022 to lines 61-62, and Clapp et al 2023 and Nugent & Bretherton 2023 to line 34 of the revised manuscript.

Frey, W., Schofield, R., Hoor, P., Kunkel, D., Ravegnani, F., Ulanovsky, A., Viciani, S., D'Amato, F., and Lane, T. P.: The impact of overshooting deep convection on local transport and mixing in the tropical upper troposphere/lower stratosphere (UTLS), Atmos. Chem. Phys., 15, 6467–6486, https://doi.org/10.5194/acp-15-6467-2015, 2015.

I had to re-read section 3.1 a number of times to form a picture of the flight path with respect to the features of the convection event. This was not helped by the fact that in figure 1, yellow and blue circles are used to mark features, but in figure 2a, points "A","B","C" and "D" are used, and then points "P1", "P2" and "P3" in figure 2b. There are only state borders marked to help relate one figure to another (I assume they are state borders, they might be something else). My suggestion would be to add a schematic diagram sketching the flight path in 3 dimensions compared to the location of the anvil and the cirrus cloud. I think something like this would be easy to do and would help the reader grasp the situation much more quickly. I found figure 5 very helpful to orient the discussion.

We appreciate the suggestion, but believe that the illustration in Figure 5 already accomplishes much of the desired detail and context here. However, we have made minor revisions to Figures 1 & 2 to better assist with visual comparison between them. Ultimately, what we hope to accomplish with Figure 1 is a review of the storms responsible for the observed composition impacts and their characteristics. Because the storms evolve quickly over time and the flight track occurs over a broad time span, we believe that there is not a single appropriate time to overlay the flight track on Figure 1. Alternatively, adding it to every map would result in unnecessary clutter atop the features we believe are important to highlight. As a compromise, we have provided an animation of satellite imagery and the ER-2 flight track as supplementary material to the revision which is referenced in lines 227-228. While the labeling in Figure 2 is somewhat complex, it is done to provide better cohesion with the aircraft imagery and trace gas analyses that follow in Figures 3 & 4, which provide much of the desired dimensionality noted here (see especially the P1-3 labels in Fig.

4a). We have also specified the latitude and longitude bounds for Figures 1 & 2 in the captions for context.

In one or two places the authors assume knowledge of US geography that an international audience might not possess - for example I had to google "Texas panhandle", and no latitudes and longitudes are given anywhere.

State labels have been added to Figure 1l and Figure 2a. The latitude and longitude bounds of the domains have been added to figure captions for Figure 1 and 2. We have rephrased the description to remove mention of the "Texas panhandle".

Line 27 I think this sentence should be re-worded, I think your meaning is that "overshooting convection" as discussed in the paper is often associated with "severe convection" in the meteorological sense and hence "severe weather", but it doesn't read that way.

Yes, that is what we mean. This sentence has been re-worded in lines 26-27 of the revised manuscript to make this clearer.

Line 36-39 I assume this statement is based on Chang et al. 2023, in which case it needs some qualification, because I would say it only applies to the overshooting events that were strong enough to be detected by the criteria used. Weaker ones wouldn't make it up as high.

This statement is based on Cooney et al 2018, and a reference to Cooney et al 2018 has been added here to clarify this on line 39.

Line 49 Define "storm-relative winds"

This definition, *"(the difference between the environmental wind and storm motion)"*, has been added to lines 51-52.

Line 50 Define (very briefly) "hydraulic jump"

See response to similar comment by Reviewer 1.

Line 122 Please give the latitude and longitude for Salina.

It has been added.

Line 171 Is 250 m uncertainty good enough for the analysis that follows?

Yes, but note response to Reviewer 1 above, which includes a figure, and revisions to the text referenced here (Section 2.3). The 250-m estimate was based on model-level analyses in Hoffmann & Spang 2022. We performed some careful comparisons between tropopauses calculated from the 37-level pressure grid data and that from the 137-level native model grid data to demonstrate that uncertainties of our tropopauses are ±500 m. Such an uncertainty is standard in global model output such as ERA5 and tolerable for the analyses and interpretations presented.

Line 187 You should give the central wavelength of these channels, not just "VIS" and "IR" and the channel numbers.

The approximate central wavelengths for VIS channel 2 (0.64 microns) and IR channel 13 (10.3 microns) have been added to line 206 of the revised manuscript.

Figure 3 I think it would help to give the direction the camera is pointing here as well as in the text. (I have to admit I'm not really seeing much in these photos).

The direction the camera is pointing for each panel has been added to the Figure 3 caption to match the text.

Lines 240-243 This isn't quite true for CH4 is it? It doesn't seem to drop between segments 1 and 2.

This sentence has been edited to state that CO and $CH_4$ are substantially elevated in segments 5 and 6, and CO is additionally elevated in segments 1 and 2.

Figure 4 Ozone is included in the plot but not discussed in the text in this section (lines 240-275).

The following sentence discussing ozone has been added to lines 259-261 of the revised manuscript. *"Potential temperature and $O_3$, stratospheric tracers, follow similar trends throughout the flight, and will be discussed in greater detail in subsequent analysis."*

Figure 5 I found this figure very helpful.

Thank you!

Line 298 I don't understand what you mean by "a H2O dominant convective signature" – do you mean it looks like there has been convection of H2O but not other tropospheric tracers?

Yes, we mean there we see transport of $H_2O$ indicating convective influence, but don't necessarily see the transport of other tropospheric tracers! The phrase "with no or

minimal impacts to other trace gases" has been added on to this statement to provide additional clarification on line 319 of the revised manuscript.

Line 325 Not "below a value of 1" but "well below", it's more like half.

This has been changed to "well below".

Lines 370-376 I find your explanation plausible, but am not convinced that it would lead to a positive correlation.

See response and edits made regarding the first minor comment by Reviewer 1.

Line 429 "desperately" seems a little bit overwrought.

"Desperately" has been deleted from this sentence.

Citation: https://doi.org/10.5194/egusphere-2023-2603-RC2